# High N-Terminal proB-Type Natriuretic Peptide Indicates Elevated Risk of Death after Percutaneous Coronary Intervention Compared to Coronary Artery Bypass Surgery in Patients with Left Ventricular Dysfunction

**DOI:** 10.3390/jcm8060898

**Published:** 2019-06-23

**Authors:** Christian Roth, Matthias Schneider, Daniel Dalos, Clemens Gangl, Christian Toth, Georg Goliasch, Rudolf Berger

**Affiliations:** 1Department of Internal Medicine II, Cardiology, Medical University of Vienna, 1090 Vienna, Austria; christian.roth@meduniwien.ac.at (C.R.); matthias.schneider@meduniwien.ac.at (M.S.); daniel.dalos@meduniwien.ac.at (D.D.); clemens.gangl@meduniwien.ac.at (C.G.); georg.goliasch@meduniwien.ac.at (G.G.); 2Department of Internal Medicine I, Cardiology and Nephrology, Hospital of St. John of God, 7000 Eisenstadt, Austria; christian.toth@bbeisen.at

**Keywords:** N-terminal proB-type natriuretic peptide, percutaneous coronary intervention, coronary-artery bypass grafting, left ventricular dysfunction, outcome

## Abstract

**Background:** Reduced left ventricular function (LVF) is a predictor for stent-thrombosis. In advanced heart failure (characterized by high NT-proBNP) with an activated coagulation system, coronary events clinically perceived as sudden death or death from heart failure may be more common in patients treated by percutaneous coronary intervention (PCI) than in patients treated by coronary artery bypass grafting (CABG). Our study analyses (1) if patients with reduced LVF who require coronary revascularization will have a better survival benefit with CABG or PCI, and (2) if the survival benefit is predicted by NT-proBNP. **Methods:** This observational retrospective study included patients from the coronary catheter laboratory database of the Medical University of Vienna (CCLD-MUW). Multivariate Cox regression analyses were performed to test the hypothesis that there is an interaction in the risk of death between those with lower or elevated NT-proBNP levels and the revascularization procedure (PCI or CABG). The relative risk of PCI compared to CABG as reference was calculated for patients with low and elevated NT-proBNP levels. **Results:** In the entire study population with 398 patients (340 PCI and 58 CABG) the revascularization procedure had no predictive value. When the revascularization procedure*NTproBNP interaction was forced into the Cox regression model, this term was an independent predictor of death. The relative risk of PCI compared to CABG was similar in patients with lower NT-proBNP—1.01 (95% confidence interval (CI), 0.45–2.24), but was significantly increased in patients with elevated NT-proBNP—1.58 (95% CI, 1.07–2.33). **Conclusion:** Death is associated to the revascularization procedure, but only in those patients with elevated NT-proBNP levels. NT-proBNP is a predicting factor for the revascularization procedure: elevated levels showed an increased risk of death after PCI compared to CABG, whereas lower levels were associated with a similar risk after both revascularization procedures.

## 1. Introduction

The first study suggesting a survival benefit of coronary artery bypass graft surgery (CABG) over medical therapy in patients with poor left ventricle function (LVF) was published in 1983 by Alderman and colleagues [1]. Recently the STICH trial (Surgical Treatment for Ischemic HF trial) confirmed this survival benefit due to CABG in patients with current guideline-based medical therapy for coronary artery disease and heart failure [2]. However, patients with poor LVF often suffer from multiple comorbidities like renal disease, chronic pulmonary disease and diabetes mellitus [3]. According to the society of thoracic surgeons (STS), score, reduced ejection fraction, symptoms of heart failure (HF) as well as various comorbidities increase the perioperative risk for CABG surgery [4]. Therefore, percutaneous coronary intervention (PCI) with drug-eluting stents (DESs) is often used for the treatment of these patients in real-world practice. However, recent guidelines do not recommend PCI to revascularize myocardium in these patients due to lack of evidence [2,5]. Few observational studies compared CABG with PCI using DESs in patients with moderate to severe left ventricular systolic dysfunction (LVSD): some of these analyses suggested comparable outcomes [6,7]; however, others showed a survival benefit by CABG compared to PCI [8,9].

Differences in the severity of HF may contribute to these conflicting results: HF is characterized by an imbalance in pro- and antithrombotic factors with clear overall prothrombotic shift [10] and this prothrombotic environment occurs predominantly in advanced HF. Natriuretic peptides are well established markers of LVSD [11] and HF [12]. As they vary from normal levels in mild stages of HF to excessive elevation in patients with advanced HF, they indicate different stages of disease [13] and predict risk of death [14]. Up to now, there is no evidence that natriuretic peptides interfere directly with clotting. Nevertheless, elevated N-terminal pro B-type natriuretic peptide (NT-proBNP) levels have been associated with spontaneous echo contrast as precursor for left atrial thrombosis [15], as well as with transcatheter heart valve thrombosis [16]. In a large cohort of chronic HF-patients NT-proBNP was an independent predictor of stroke [17]. These findings indicate that elevated NT-proBNP levels are related with thrombosis. However, this association is probably not based on a direct influence of natriuretic peptides on the coagulation system, but reflects a parallel activation of natriuretic peptides and the coagulation system.

Acute coronary findings are frequent and usually not clinically diagnosed in HF-patients with coronary artery disease (CAD), in those dying suddenly as well as in those dying from pump failure [18,19]. Coronary events are thus suggested to be main triggers for these two modes of death. Stent-implantation during PCI generally involves the risk of stent thrombosis and may significantly increase the risk to suffer further coronary events especially in patients with advanced heart failure and, thereby, activated coagulation system: indeed, the Dutch Stent Thrombosis Registry reported that reduced LVEF is a predictor for stent-thrombosis [20]. These findings suggest that in advanced heart failure (characterized by high NT-proBNP), coronary events clinically perceived as sudden death or death from heart failure are more common in PCI patients than in CABG patients. In mild stages of heart failure (characterized by low NT-proBNP) where there is no prothrombotic milieu, the frequency of coronary and related clinical events may be equally distributed in PCI and CABG patients. Our study analyses (1) if patients with reduced LVF who require coronary revascularization will have a better survival benefit with CABG or PCI, and (2) if the survival benefit is predicted by NT-proBNP.

## 2. Methods

### 2.1. Study Population

This single-center observational retrospective cohort study selected patients from the coronary catheter laboratory database of the Medical University of Vienna (CCLD-MUW) between the years 2004 and 2012 who suffered from significant CAD and moderate to severe LVSD. Significant CAD was defined as diameter stenosis for epicardial coronary artery ≥70% or 50% to 70% with objective evidence of myocardial ischemia and for left main coronary artery ≥50% and had to be present in at least one major epicardial coronary artery. Experienced interventionists approved all coronary angiograms. CABG or PCI were performed according to contemporary guidelines. PCI was usually the preferred option for one—or two-vessel disease not involving the proximal LAD. Especially in patients with complex coronary artery disease (CAD), the decision was mostly reached utilizing a Heart Team approach, which included the opinion of an interventional cardiologist and a cardiovascular surgeon. Often both procedures were possible options and the final decision was that of the patient and his/her advocate physician in the context of the overall clinical presentation. Complex coronary artery disease with a SYNTAX-score (an angiographic tool for grading the complexity of coronary artery disease) >22 was preferably treated by CABG. The PCI was performed almost exclusively within the hospital stay in which the diagnostic coronary angiography took place. CABG was usually performed within three months after the diagnostic coronary angiography (median waiting time 14 days; IQR 6 to 43 days). The patients were followed up starting with the initial coronary angiography up to the due day (31 December 2014). LVSD was determined by echocardiography and had to be characterized by an ejection fraction below 40% (Simpson method) or visual estimation (moderately, moderate to severely, and severely reduced). For eligible patients NT-proBNP before revascularization had to be available. The study is in line with the Declaration of Helsinki and was approved by the ethics committee of the Medical University of Vienna (EC number: 972/2011).

### 2.2. Data Collection

The CCLD-MUW is a comprehensive database, which had been established in co-operation with the IT department of the MUW. Study data including baseline characteristics, co-morbidities, angiographic characteristics, echocardiographic parameter including LV function and blood results. The survival status including date and type of death was recorded for each patient up to the due date with a maximum observation period of up to ten years. On the due date, this survival status was retrieved from the Austrian death registry database.

### 2.3. Statistical Analysis

Continuous variables are presented as mean ± SD or median with interquartile range (IQR) as appropriate, categorical variables as counts or percentages as appropriate. Continuous variables were compared using an unpaired T-test, categorical data were compared using crosstabs and chi-square test. To identify independent predictors of death from any cause, the Cox proportional hazards model with stepwise backward elimination methods (retention threshold, *p* < 0.05) was used to adjust potential confounders identified by the investigators using literature search and based on clinical knowledge. As NT-proBNP levels were not normally distributed, log transformed NT-proBNP levels were used for continuous analysis within the Cox proportional hazards regression models. No method was used to impute missing values or adjust the model for the presence of missing data. Clinically relevant covariates were introduced into various multivariable models:-To answer if there is an association between type of revascularization procedure and death, death was adjusted by age, sex, rhythm, prior myocardial infarction, diabetes mellitus (DM), estimated glomerular filtration rate (Cockcroft-Gault formula; eGFR), New York Heart Association (NYHA) class, Canadian Cardiovascular Society (CCS) class, number of coronary vessels diseased and revascularization procedure in Model A.-To answer if death is associated by levels of NT-proBNP, death was adjusted by age, sex, rhythm, prior myocardial infarction, DM, eGFR, NYHA class, CCS class, number of coronary vessels diseased and NT-proBNP in Model B.-To answer if there is an interaction between revascularization procedure and NT-proBNP, death was adjusted by age, sex, rhythm, prior myocardial infarction, DM, eGFR, NYHA class, CCS class, number of coronary vessels diseased, NT-proBNP, revascularization procedure, and NT-proBNP*revascularization procedure in Model C.-To confirm an association between type of revascularization procedure and death in patients with above median NT-proBNP, death was adjusted by age, sex, rhythm, prior myocardial infarction, DM, eGFR, NYHA class, CCS class, number of coronary vessels diseased, and revascularization procedure in Model D.-To confirm no association between type of revascularization procedure and death in patients with below median NT-proBNP, death was adjusted by age, sex, rhythm, prior myocardial infarction, DM, eGFR, NYHA class, CCS class, number of coronary vessels diseased, and revascularization procedure in Model E.

Variables which were significant in these COX regressions were displayed in Forest plots. The rule of thumb that Cox models should be used with a minimum of ten outcome events per predictor variable applies to the entire study population (173 deaths) and to patients with NT-proBNP above the median (123 deaths). This rule is not exactly met in patients with NT-proBNP below the median (50 deaths). While this rule was discussed as too conservative [21], we recalculated the Cox Model E using only univariate significant confounders (age, eGFR, number of coronary vessels diseased, DM) and the type of revascularization procedure to confirm the accuracy of this model. Bivariate models included NT-proBNP, one of the available parameters of thrombosis (D-Dimer) or coagulation (thromboplastin time, partial thromboplastin time, International Normalized Ratio) and an according interaction term (NT-proBNP*parameter of thrombosis or coagulation) and assessed if thrombosis is a mediator in the association between death and levels of NT-proBNP. The NT-proBNP median (3042 pg/mL; interquartile range (IQR): 1313 to 8473 pg/ml) was used as cutoff point to differentiate lower and elevated values. As there was an even number of NT-proBNP values in the data set, the median was found by taking the mean (average) of the two middlemost numbers. The relative risk of PCI compared to CABG as reference was calculated for patients with low and elevated NT-proBNP levels using a contingency table. Statistical analyses were performed using the SPSS software, version 24.0.

## 3. Results

A total of 398 patients with moderate to severe LVSD and CAD were selected from the CCLD-MUW. The detailed flow chart is presented in Figure 1. All patients’ characteristics are listed in Table 1 and Table 2. During a median observation period of 1495 days (IQR 819/2458) 173 patients (44%) died. The causes of death are given in Table 3.

### 3.1. Results of the Cox Proportional Hazards Regression Models

Figure 2 shows independent predictors of death as calculated in different Cox proportional hazards models: as evaluated in Model A, the type of revascularization procedure was not an independent predictor of death in the entire study population including 398 patients (340 PCI and 58 CABG). The outstanding importance of NT-proBNP as a prognostic marker in patients with reduced LVF was confirmed in Model B: NT-proBNP was the strongest independent predictor of death, followed by age, and eGFR. Model C demonstrated a significant interaction between NT-proBNP and the revascularization procedure: death was associated to the revascularization procedure, but only in those patients with elevated NT-proBNP levels.

To evaluate the impact of the revascularization procedure on the risk of death in different stages of HF, patients were stratified according to their NT-proBNP median (3042 pg/mL): Out of 199 patients with above median levels 161 patients underwent PCI and 38 patients had CABG surgery. Out of 199 patients with below median levels 179 patients were treated with PCI and 20 patients underwent CABG surgery. Model D confirmed an association between the type of revascularization procedure and death in patients with above median NT-proBNP, whereas this association could not be demonstrated in patients with NT-proBNP levels below the median (Model E). When recalculating this model using only univariate significant confounders (age, eGFR, number of coronary vessels diseased, DM) and the type of revascularization procedure, the result stayed the same (DM as the only independent predictor of death).

Using bivariate Cox regression models, we found no interactions between NT-proBNP and one of the available parameters for thrombosis (D-Dimer) or coagulation (thromboplastin time, partial thromboplastin time, International Normalized Ratio).

### 3.2. Relative Risk of Death of PCI Compared to CABG in Patients with Lower and Elevated NT-proBNP

The relative risk of death in patients having a PCI compared to CABG was similar in patients with lower NT-proBNP—1.01 (95% CI, 0.45–2.24), but was significantly increased in patients with elevated NT-proBNP—1.58 (95% CI, 1.07–2.33).

## 4. Discussion

Death is associated to the revascularization procedure, but only in those patients with reduced LVF and elevated NT-proBNP levels. NT-proBNP is a predicting factor for the revascularization procedure: elevated levels showed an increased risk of death after PCI compared to CABG, whereas lower levels were associated with a similar risk after both revascularization procedures.

### 4.1. Comparison of Outcome after PCI and CABG

There is a lack of randomized controlled trials testing PCI versus medical therapy in patients with LVSD. Trials comparing PCI versus CABG have routinely excluded patients with severe LVSD. Recently a large meta-analysis involved 8782 HF-patients with reduced ejection fraction of 16 studies (published between 1983 and 2015) including 2 randomized controlled trials. This analysis demonstrated a reduction in mortality with CABG compared with PCI [9]. Two recently published registries analyzed survival after CABG compared with PCI with conflicting results [6,8]. A merged database of 3 large registries included 442 patients after CABG and 469 patients after PCI and found a lower risk of all-cause death with CABG after adjustment for multiple confounders [8]. At the same time, a large registry included 4616 patients and analyzed 2126 patients with similar propensity scores. This registry found no difference in long-term survival between PCI and CABG [6]. Our study contributes to the understanding of these divergent outcomes by showing that the increasing severity of HF reflected by elevated NT-proBNP can increase the risk of death after PCI compared to CABG. On the other hand, less severe HF as reflected by lower NT-proBNP levels is associated with a similar risk after both revascularization procedures. None of these two previous analyses reported parameters describing the severity of HF like natriuretic peptide levels or NYHA class [6,8].

### 4.2. Mode of Death

Myocardial infarction is suggested to be a significant trigger for both sudden death as well as pump failure death in patients with HF and significant CAD [18,19]. In these patients coronary findings like ruptured plaques, coronary thrombus or myocardial infarction are frequent and usually not clinically diagnosed: the ATLAS trial [18] evaluated the effects of two different lisinopril doses in 3164 patients with New York Heart Association class II to IV HF and an ejection fraction < 30%. Autopsy results were reported in 171 patients who died (12%). The percentage of patients classified as dying of MI was 28% (48 of 171) in the autopsy group versus 4% (7 of 171) in the non-autopsied group. In patients with significant CAD an acute coronary finding was described in 54% of patients who died suddenly, and in 32% of patients who died from pump failure. In contrast, in non-CAD patients, acute coronary findings were present in only 5% and 10% respectively. The OPTIMAAL trial [19] randomized 5477 patients with LVSD (LVEF > 35%) or HF following an acute myocardial infarction to losartan or captopril. An autopsy report was available from 180 patients who died during follow-up (19%). Acute MI was found in 57% (102 of 180) of the autopsies, but was diagnosed by an endpoints adjudication committee using clinical data in only 29 cases (16%). In detail, an acute MI was found at autopsy in 55% (37 of 67) of the deaths that had been classified as due to an arrhythmia and in 81% (21 of 26) of the deaths classified as due to pump failure.

In our study cohort myocardial infarction also was rarely reported as cause of death. The following findings suggest myocardial infarction to be significantly undetected: (I) severe HF-patients treated with PCI had a worse prognosis than those treated with CABG; (II) severe HF-patients were characterized by a more pronounced prothrombotic state than patients with a milder stage of HF.

### 4.3. Prothrombotic State and Inflammation

In HF the coagulation system is characterized by a prothrombotic shift [19] with increased levels of plasma coagulation factors (plasma fibrinogen, fibrinopeptide A), markers of fibrinolysis (D-dimer) and markers of platelet activity [10,22]. Our analysis refines this knowledge by the finding that the prothrombotic activation in HF depends on the severity of the disease as defined by natriuretic peptide activation: patients with above median NT-proBNP levels in our study were characterized by higher levels of fibrinogen and D-dimer compared to patients with below-median levels. Similarly, in a study of older British men with no prevalent arterial or venous thrombosis D-dimer and vWF were significantly associated with NT-proBNP [23]. In contrast to venous clots which are fibrin-rich and formed during lower shear stress environments, stent-thrombosis in coronary arteries is mainly mediated by platelets activation. Platelet activation due to heart failure may potentiate the risk of stent thrombosis. As platelet function testing is not routinely performed, measures for platelet activation are not available in our patients. Available parameters of thrombosis (D-Dimer) or coagulation (thromboplastin time, partial thromboplastin time, International Normalized Ratio) did not demonstrate interactions with NT-proBNP*revascularization procedure. This may be due to the fact that changes in plasma coagulation are not mainly involved in the development of atherothrombosis. In addition, the number of patients with available parameters of thrombosis and coagulation is limited.

### 4.4. Limitations

In our study we used Cox proportional hazards models to identify independent predictors of death and to adjust for potential confounders. As our study is of observational nature, we were not able to correct for unmeasured variables. For the same reason, a direct comparison of patients with PCI and CABG is difficult: the selection of treatment was influenced by patient characteristics and patient and physician preferences. In particular, patients with less extensive disease, especially those with a one-vessel disease, were preferably treated with PCI instead of CABG. However, in the Cox model the issue of single vessel versus multivessel coronary artery disease had no impact on survival.

## 5. Conclusions

In patients with moderate to severe LVSD and CAD, elevated NT-proBNP indicates an increased risk of death after PCI compared to CABG. Further studies should evaluate if higher rates of stent-thrombosis may account for these results. Newer anticoagulants could not improve survival rate in the treatment of patients with LVSD due to CAD [24]. However, the extension of dual anti-plateled therapy may be a better way to improve survival in these patients [25]. Large outcome studies are needed to test this hypothesis.

## Figures and Tables

**Figure 1 jcm-08-00898-f001:**
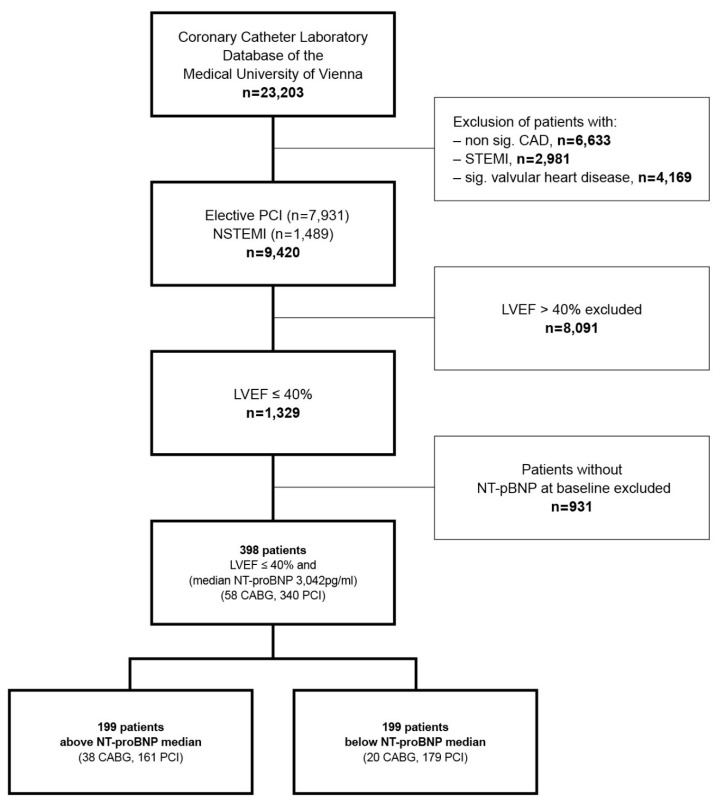
Flow chart. CABG = coronary artery bypass graft, CAD = coronary artery disease, DES = drug eluting stent, LVEF = left ventricle ejection fraction, NSTEMI = non-ST-segment elevation myocardial infarction, NT-proBNP = N-terminal proB-type natriuretic peptide, PCI = percutaneous coronary intervention, STEMI = ST-segment elevation myocardial infarction.

**Figure 2 jcm-08-00898-f002:**
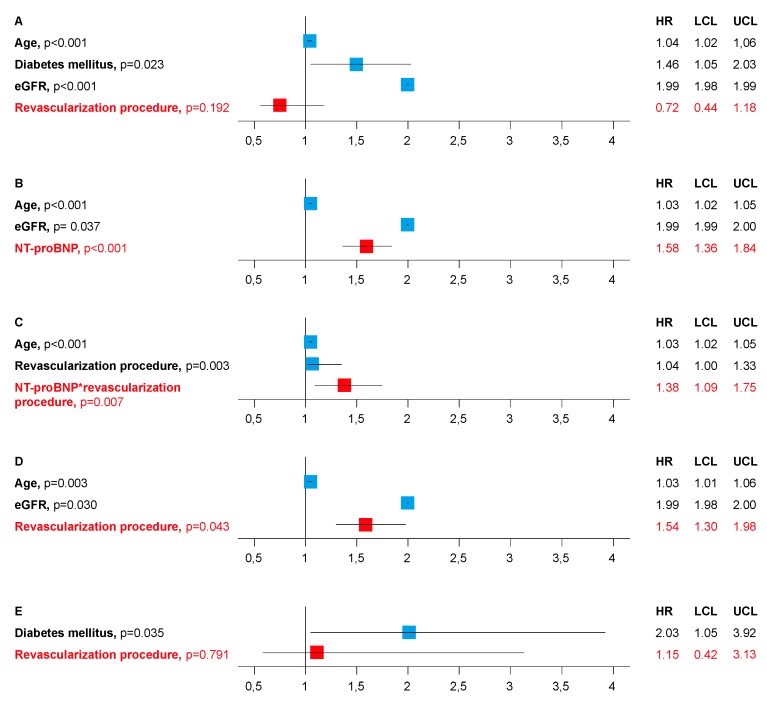
Predictors of death in Cox proportional hazards regression analyses. (**A**) The type of revascularization procedure was not an independent predictor of death in the entire study population. (**B**) NT-proBNP was the strongest independent predictor of death. (**C**) A significant interaction between NT-proBNP and the revascularization procedure: death was associated to the revascularization procedure, but only in those patients with elevated NT-proBNP levels. (**D**) Association between the type of revascularization procedure and death in patients with above median NT-proBNP. (**E**) No association between the type of revascularization procedure and death in patients with below median NT-proBNP.

**Table 1 jcm-08-00898-t001:** Demographics, comorbidities and clinical presentation of all patients, subgroups with above and below NT-pBNP median (3042 pg/mL, interquartile range (IQR) 1313/8473), and subgroups according to the type of revascularization procedure.

	All *n* = 398	Above NT-pBNP Median (>3042 pg/mL) *n* = 199	Below NT-pBNP Median (<3042 pg/mL) *n* = 199	*p*-Value	Revascularization Procedure		*p*-Value
					PCI *n* = 340	CABG *n* = 58	
**Age, years; median (IQR)**	67 (59/75) *	70 (64/78) *	64 (55/71) *	<0.001	66 (58/76)	69 (61/73)	0.404
**Gender**				0.526			0.661
**Female; *n* (%)**	77 (19)	41 (21)	36 (18)	-	67 (20)	10 (17)	-
**Male; *n* (%)**	321 (81)	158 (7 9)	163 (82)	-	273 (80)	48 (83)	-
**Body-Mass-Index, kg/m^2^; median (IQR)**	27 (25/30)	26 (24/29)	28 (25/30)	0.017	27 (25/30)	27 (25/31)	0.708
**Hypertension; *n* (%)**	306 (77)	156 (78)	150 (75)	0.476	261 (77)	45 (78)	0.819
**Hyperlipidemia; *n* (%)**	295 (74)	143 (72)	152 (76)	0.303	254 (75)	41 (71)	0.519
**Diabetes mellitus; *n* (%)**	168 (42) *	95 (48)	73 (37) ^†^	0.026	133 (39)	35 (60)	0.002
**Smoker**				0.021			0.850
**Current; *n* (%)**	118 (30)	53 (27)	65 (33)	-	102 (30)	16 (28)	-
**Previous; *n* (%)**	79 (20)	32 (16)	47 (24)	-	66 (19)	13 (22)	-
**Family history; *n* (%)**	76 (19)	31 (16)	45 (23)	0.074	67 (20)	9 (16)	0.453
**Cerebrovascular and/or peripheral vascular disease; *n* (%)**	124 (31) *	86 (43)	38 (19) ^†^	<0.001	97 (29)	27 (47)	0.006
**Prior myocardial infarction; *n* (%)**	155 (39) ^‡^	76 (38) ^‡^	79 (40)	0.758	135 (45)	20 (35)	0.451
**Prior PCI; *n* (%)**	94 (24)	38 (19)	56 (28)	0.034	80 (24)	14 (24)	0.920
**Prior CABG; *n* (%)**	54 (14)	21 (11) ^‡^	31 (16)	0.242	53 (16)	1 (2)	0.004
**NYHA class**	^†^			0.076			0.800
**Class I; *n* (%)**	81 (23)	40 (22)	41 (24)	-	69 (22)	12 (23)	-
**Class II; *n* (%)**	55 (15)	21 (11)	34 (20)	-	48 (16)	7 (14)	-
**Class III; *n* (%)**	168 (47)	91 (49)	77 (44)	-	146 (47)	22 (43)	-
**Class IV; *n* (%)**	55 (15)	34 (18)	21 (12)	-	45 (15)	10 (20)	-
**CCS class**	^‡^			0.252			0.303
**Class I; *n* (%)**	61 (17)	36 (19)	25 (15)	-	51 (17)	10 (20)	-
**Class II; *n* (%)**	77 (21)	33 (18)	44 (25)	-	69 (22)	8 (16)	-
**Class III; *n* (%)**	127 (36)	69 (37)	58 (34)	-	112 (36)	15 (29)	-
**Class IV; *n* (%)**	94 (26)	48 (26)	46 (27)	-	76 (25)	18 (35)	-
**Elective/NSTEMI**				0.128			0.915
**Elective; *n* (%)**	169 (43)	77 (39)	92 (46)	-	144 (42)	25 (43)	-
**NSTEMI; *n* (%)**	229 (57)	122 (61)	107 (54)	-	196 (58)	33 (57)	-

Univariate predictor for death: * = *p* < 0.001; ^†^ = *p* ≤ 0.010; ^‡^ = *p* ≤ 0.050. CABG = coronary artery bypass graft, CCS = Canadian Cardiovascular Society, IQR = interquartile range, MV = multi-vessel disease, NSTEMI = non-ST-segment elevation myocardial infarction, NYHA = New York Heart Association, PCI = percutaneous coronary intervention.

**Table 2 jcm-08-00898-t002:** Baseline therapy and electrocardiographic, echocardiographic and coronary angiographic results of all patients, subgroups with above and below NT-pBNP median (3042 pg/mL, IQR 1313/8473), and subgroups according to the type of revascularization procedure.

	All *n* = 398	Above NT-pBNP Median (>3042 pg/mL) *n* = 199	Below NT-pBNP Median (<3042 pg/mL) *n* = 199	*p*-Value	Revascularization Procedure		*p*-Value
					PCI *n* = 340	CABG *n* = 58	
**ACE-inhibitor/ARB; *n* (%)**	398 (100)	199 (100)	199 (100)	-	340 (100)	58 (100)	-
**Beta-Bocker; *n* (%)**	316 (79)	156 (78)	160 (80)	0.620	271 (80)	45 (78)	0.712
**Aldosteron; *n* (%)**	198 (50)	106 (53)	92 (46)	0.160	168 (49)	26 (45)	0.417
**Aspirin; *n* (%)**	398 (100)	199 (100)	199 (100)	-	340 (100)	58 (100)	-
**Clopidogrel; *n* (%)**	315 (79)	152 (76)	163 (82)	0.175	315 (93)	-	-
**Ticagrelor; *n* (%)**	25 (6)	9 (5)	16 (8)	0.148	25 (7)	-	-
**Pacemaker, *n* (%)**	29 (7)	18 (9)	11 (6)	0.177	28 (8)	1 (2)	0.078
**ICD, *n* (%)**	46 (12)	18 (9)	28 (14)	0.117	42 (12)	4 (7)	0.230
**NT-proBNP, pg/ml (*n*); median (IQR)**	3042 (1313/8473) *	8447 (5195/15,330) *	1313 (579/2116) ^‡^	<0.001	2927 (1282/8524)	4280 (1624/8533)	0.392
**Total cholesterol, mg/dl; median (IQR)**	175 (144/211) ^†^	167 (140/206)	181 (150/217)	0.003	175 (144/213)	175 (142/205)	0.564
**HDL, mg/dl; median (IQR)**	42 (34/49)	39 (33/48)	43 (34/51)	0.052	42 (35/50)	40 (32/49)	0.438
**LDL, mg/dl; median (IQR)**	93 (72/116)	92 (72/116)	96 (71/116)	0.814	89 (69/113)	109 (91/168)	0.076
**Creatine, mg/dl; median (IQR)**	1.13 (0.94/1.47)	1.28 (1.00/1.74)	1.06 (0.91/1.24)	<0.001	1.12 (0.94/1.46)	1.18 (0.93/1.47)	0.416
**eGFR, median (IQR)**	68 (49/93) *	55 (35/80) ^‡^	81 (61/102) ^†^	<0.001	66 (48/94)	70 (51/89)	0.432
**Sodium, mmol/L; median (IQR)**	139 (137/140)	138 (136/141)	139 (137/140)	0.945	139 (137/141)	138 (137/140)	0.974
**Potassium, mmol/L; median (IQR)**	4.1 (3.8/4.3)	4.1 (3.7/4.4)	4.1 (3.8/4.3)	0.679	4.1 (3.8/4.4)	4.1 (3.9/4.3)	0.264
**CRP, mg/dl; median (IQR)**	1.35 (0.48/4.77)	2.43 (0.98/9.21)	0.72 (0.35/2.17)	<0.001	1.41 (0.45/4.79)	1.31 (0.54/3.65)	0.706
**D-Dimer, µg/ml; median (IQR)**	1.03 (0.45/1.95)	1.40 (0.86/2.49)	0.54 (0.34/1.11) *	0.004	0.95 (0.41/1.83)	1.69 (0.86/2.61)	0.200
**Fibrinogen mg/dl; median (IQR)**	482 (396/579) *	525 (440/642)	451 (376/537)	<0.001	481 (394/580)	517 (424/578)	0.430
**Prothrombine time, %; median (IQR)**	90 (75/107)	87 (72/104)	93 (78/108)	0.012	90 (75/106)	94 (78/112)	0.253
**aPTT, s; median (IQR)**	37 (33/45)	39 (34/46)	36 (33/42)	0.138	38 (34/44)	37 (33/45)	0.771
**INR; median (IQR)**	1.2 (1.1/1.4)	1.4 (0.9/2.5)	1.2 (1.1/1.3)	0.052	1.2 (1.1/1.4)	1.2 (1.1/1.3)	0.988
**Heart rate, bpm; median (IQR)**	77 (66/88)	80 (69/90)	74 (64/84)	<0.001	77 (66/88)	73 (66/86)	0.999
**Rhythm**	^‡^			0.016			0.641
**Sinus rhythm; *n* (%)**	345 (87)	162 (81)	183 (92)	-	292 (86)	53 (91)	-
**Atrial fibrilation; *n* (%)**	35 (9)	24 (12)	11 (6)	-	34 (9)	4 (7)	-
**Other; *n* (%)**	18 (4)	13 (7)	5 (2)	-	17 (5)	1 (2)	-
**Left ventricular function, systolic**	*			0.017			0.983
**Moderately impaired; *n* (%)**	148 (37)	61 (31)	87 (44)	-	127 (37)	21 (36)	-
**Moderately to severley impaired; *n* (%)**	130 (33)	68 (34)	62 (31)	-	111 (33)	19 (33)	-
**Severely impaired; *n* (%)**	120 (30)	70 (35)	50 (25)	-	102 (30)	18 (33)	-
**No. of coronary vessels diseased**	*		^‡^	0.031			<0.001
**1 VD; *n* (%)**	112 (28)	46 (23)	66 (33)	-	112 (33)	-	-
**2 VD; *n* (%)**	83 (21)	39 (20)	44 (22)	-	72 (21)	11 (19)	-
**3 VD; *n* (%)**	203 (51)	114 (57)	89 (45)	-	156 (46)	47 (81)	-
**Revascularization procedure**		^‡^		0.011			
**PCI; *n* (%)**	340 (85)	161 (81)	179 (90)				
**CABG; *n* (%)**	58 (15)	38 (19)	20 (10)				

Univariate predictor for death: * = *p* < 0.001; ^†^ = *p* ≤ 0.010; ^‡^ = *p* ≤ 0.050. ACE = angiotensin-converting-enzyme, aPTT = activated partial thromboplastin time, ARB = angiotensin-receptor blocker, CABG = coronary artery bypass graft, CRP = C-reactive protein, eGFR = estimated glomerular filtration rate (Cockcroft-Gault formula), HDL = high density lipoprotein, ICD = implantable cardioverter-defibrillator, INR = International Normalized Ratio, IQR = interquartile range, LDL = low density lipoprotein, NSTEMI = non-ST-segment elevation myocardial infarction, NT-proBNP = N-terminal proB-type natriuretic peptide, PCI = percutaneous coronary intervention, VD = vessel disease.

**Table 3 jcm-08-00898-t003:** Cause of death.

All	All *n* = 398	CABG *n* = 58	PCI-All *n* = 340	*p*-Value CABG vs. PCI-All
**Total death; n (%)**	173 (44)	21 (36)	152 (45)	0.227
**- Sudden death; *n* (%)**	16 (4)	2 (3)	14 (4)	-
**- Heart failure; *n* (%)**	99 (25)	14 (24)	85 (25)	-
**- Myocardial infarction; *n* (%)**	30 (8)	1 (2)	29 (9)	-
**- Other; *n* (%)**	28 (7)	4 (7)	24 (7)	-
**Alive; *n* (%)**	225 (56)	37 (64)	188 (55)	-
**Above NT-pBNP median**	**All** ***n* = 199**	**CABG** ***n* = 38**	**PCI-All** ***n* = 161**	
**Total death; *n* (%)**	123 (62)	16 (45)	107 (66)	0.005
**- Sudden death; *n* (%)**	14 (7)	2 (5)	12 (8)	-
**- Heart failure; *n* (%)**	70 (36)	10 (26)	60 (37)	-
**- Myocardial infarction; *n* (%)**	21 (10)	1 (3)	20 (12)	-
**- Other; *n* (%)**	18 (9)	3 (8)	15 (9)	-
**Alive; n (%)**	76 (38)	22 (58)	54 (34)	-
**Below NT-pBNP median**	**All** ***n* = 199**	**CABG** ***n* = 20**	**PCI-All** ***n* = 179**	
**Total death; *n* (%)**	50 (25)	5 (25)	45 (25)	0.989
**- Sudden death; *n* (%)**	2 (1)	0 (0)	2 (1)	-
**- Heart failure; *n* (%)**	29 (15)	4 (20)	25 (14)	-
**- Myocardial infarction; *n* (%)**	9 (5)	0 (0)	9 (5)	-
**- Other; *n* (%)**	10 (5)	1 (5)	9 (5)	-
**Alive; *n* (%)**	149 (75)	15 (75)	134 (75)	-

CABG = coronary artery bypass graft, NT-proBNP = N-terminal proB-type natriuretic peptide; PCI = percutaneous coronary intervention.

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
