# Peer review of "High N-Terminal proB-Type Natriuretic Peptide Indicates Elevated Risk of Death after Percutaneous Coronary Intervention Compared to Coronary Artery Bypass Surgery in Patients with Left Ventricular Dysfunction"

_jcm, 2019, doi:10.3390/jcm8060898_

Round 1
Reviewer 1 Report
Introduction
In this sentence “these 37 patients often suffer from multiple comorbidities” do the authors mean patients with poor left ventricular function? Please make this clear.
In this line “suggested comparable outcomes [6, 7], however, other analyses showed a survival benefit by CABG 45 compared to PCI [8, 9].” I also assume that this is in patients with poor ventricular function?
This reviewer suggest that the authors make a mechanism driven hypothesis in the association between NT-proBNP and thrombosis. Are higher levels of NT-proBNP positively associated with thrombosis because NT-proBNP is higher in more severe cases of HF? In that case NT-proBNP would be not be on the causal pathway.
Which of these two situations are the authors describing?
HF increase NT-proBNP increase thrombosis death
Or
Increase NT-proBNP
HF Death
Increase thrombosis
This introduction fails to explain or propose an explanation why it would make a difference in mortality when choosing one surgical procedure over the other in patients with low or higher levels of NTproBNP.
The hypothesis in this study is to assess if NT-proBNP is a mediating factor predicting mortality between patients with LVSD who have undergone PCI vs. CABG.
Be specific about outcome. The authors should mention if they are studying morbidity or mortality or both. Be specific with regards to the outcome, length of stay, post-surgery complications, death…
Statistical analysis.
The statistical analysis does not assess the hypothesis that there is an interaction in outcome of interest between those with low or elevated levels of NT-proBNP and PCI or CABG
A contingency table should be made
NTproBNP | ||
low normal | elevated | |
PCI | mortality | mortality |
CABG | mortality | mortality |
With that model you calculate RR for each group and the SURGICAL PROCEDURE X NTproBNP interaction. That analysis should be able to tell you if the higher mortality in those with greater values of NT-proBNP is different by surgical procedure.
Analyzing the data this way will give you a larger sample size than breaking the analysis into two smaller populations
Are D-dimer values a measure of thrombosis, do you have any of the coagulation parameters for example: INR?
A model with and without D-dimer or any of the coagulation parameters should be performed to assess if thrombosis is a mediator in the association between outcome and levels of NT-proBNP and also perform an interaction model using D-dimer with NT-proBNP and also by surgical procedure.
Why was a value of 199 NT-proBNP chosen? I assume that the value is ≥ and not just >?
In this sentence “Those variables, which showed statistically 92 significant influence” substitute “influence” with “association”.
Results
Figures 2 and 3 should be replaced in a different format. The current format makes the forest plots very small and difficult to view. In addition, the authors are repeating information from figure 1. It would be easier to read if figures 2A, 2B and 2C (same for 3A, 3B and 3C) would be shown separately with appropriate titles and figure legends of their own.
Which of the surgical procedures is the reference, PCI or CABG?
Discussion.
This cannot be stated accurately “Patients with moderate to severe LVSD and CAD who have a lower NT-proBNP level have a 173 long-term survival comparable to CABG after PCI” The appropriate statistical procedures was not done to demonstrate differences in mortality by surgical procedure.
Typos or grammar issues
Older studies is not very descriptive, please indicate a time range or related to pre or after a certain study was published or procedure was accepted
Line 51 Insert comma after “recently.”
“Recently, elevated N-terminal proB-51 type natriuretic peptide (NT-proBNP) levels…”
Author Response
Dear Dr. Zhang,
herewith, we would like to resubmit our manuscript “High N-terminal proB-type Natriuretic Peptide Indicates Elevated Risk of Death After Percutaneous Coronary Intervention Compared to Coronary Artery Bypass Surgery in Patients With Left Ventricular Dysfunction” (Manuscript ID: jcm-491605) following its revision according to the issues raised by the reviewers. In the following, we respond point-by-point to the comments and describe the consequent changes made to the manuscript and the tables. The changes in the manuscript were highlighted in yellow. All authors have read and approved submission of the manuscript. We believe that the adaptations according to the reviewers´ suggestions have significantly improved the manuscript and hope you will concur with this view.
Sincerely yours,
Rudolf Berger
Rudolf Berger, MD, MSc
rudolf.berger@bbeisen.at
Johannes von Gott-Platz 1
7000 Eisenstadt
REVIEWER 1:
Introduction:
1. Line 37: in this sentence “these patients often suffer from multiple comorbidities” do the authors mean patients with poor left ventricular function? Please make this clear.
We clarified this sentence as follows:
“However, patients with poor LVF often suffer from multiple comorbidities like renal disease, chronic pulmonary disease and diabetes mellitus [3].”
See: page 2, introduction, paragraph 1
2. Line 45: in this line “suggested comparable outcomes [6, 7], however, other analyses showed a survival benefit by CABG compared to PCI [8, 9].” I also assume that this is in patients with poor ventricular function?
All these analyses evaluated revascularization in patients with moderate to severe left ventricular dysfunction. We changed this section accordingly:
“Few observational studies compared CABG with PCI using DESs in patients with moderate to severe left ventricular systolic dysfunction (LVSD): some of these analyses suggested comparable outcomes [6, 7], however, others showed a survival benefit by CABG compared to PCI [8, 9].”
See: page 2, introduction, paragraph 1
3. This reviewer suggests that the authors make a mechanism driven hypothesis in the association between NT-proBNP and thrombosis. Are higher levels of NT-proBNP positively associated with thrombosis because NT-proBNP is higher in more severe cases of HF? In that case NT-proBNP would not be on the causal pathway.
Which of these two situations are the authors describing?
HF - increase NT-proBNP - increase thrombosis - death
OR
- Increase NTproBNP -
HF Death
- Increase thrombosis -
We describe the second situation and clarified this as follows:
“Accordingly, NT-proBNP is positively associated with thrombosis because NT-proBNP is higher in more severe cases of heart failure and, therefore, reflects higher activation of a prothrombotic environment.”
See: page 2, introduction, paragraph 2
4. This introduction fails to explain or propose an explanation why it would make a difference in mortality when choosing one surgical procedure over the other in patients with low or higher levels of NTproBNP.
We explain why it would make a difference in mortality when choosing one procedure over the other in patients with low or higher levels of NTproBNP as follows:
“Therefore, patients with moderate to severe LVSD and elevated NT-proBNP levels may have an increased risk of stent-thrombosis and accordingly an increased risk of death. Indeed, the Dutch Stent Thrombosis Registry reported that reduced LVEF is a predictor for stent-thrombosis [18].”
See: page 2, introduction, paragraph 3
5. The hypothesis in this study is to assess if NT-proBNP is a mediating factor predicting mortality between patients with LVSD who have undergone PCI vs. CABG.
We changed the background-section in the abstract and the introduction accordingly:
“This study analyses if N-terminal proB-type natriuretic peptide (NT-proBNP) is a mediating factor predicting mortality in patients with left ventricular systolic dysfunction (LVSD) and coronary artery disease (CAD) who have undergone percutaneous coronary intervention (PCI) compared to coronary artery bypass grafting (CABG).”
See: page 1, abstract, background
“Our study analyses if NT-proBNP is a mediating factor predicting mortality in patients with LVSD and CAD who have undergone PCI compared to CABG.”
See: page 2, introduction, paragraph 2
6. Be specific about outcome. The authors should mention if they are studying morbidity or mortality or both. Be specific with regards to the outcome, length of stay, post-surgery complications, death…
We specified the term “outcome” in the following sequences:
Title:
“High N-terminal proB-type Natriuretic Peptide Indicates Elevated Risk of Death After Percutaneous Coronary Intervention Compared to Coronary Artery Bypass Surgery in Patients With Left Ventricular Dysfunction”
See: page 1, title
Abstract, results:
“When the mode of revascularization*NTproBNP interaction was forced into the Cox regression model, this term was an independent predictor of death. The relative risk of PCI compared to CABG was similar in patients with lower NT-proBNP - 1.01 (95% confidence interval [CI], 0.45–2.24), but was significantly increased in patients with elevated NT-proBNP – 1.53 (95% CI, 1.05–2.22). These results were confirmed in a more homogenous study population after excluding patients with single-vessel disease (multi-vessel disease - 228 PCI and 58 CABG).”
See: page 1, abstract, results
Abstract, conclusion:
“Our results confirm the hypothesis that there is an interaction in the risk of death between those with lower or elevated NT-proBNP levels and the mode of revascularization (PCI or CABG). The relative risk of PCI compared to CABG was similar in patients with lower NT-proBNP, but was significantly increased in patients with elevated NT-proBNP.”
See: page 1, abstract, conclusion
Introduction:
“Natriuretic peptides vary from normal levels to excessive elevation in patients with moderate to severe LVSD and indicate different stages of disease [11] and predict risk of death [12].”
See: page 2, introduction, paragraph 2
“Therefore, patients with moderate to severe LVSD and elevated NT-proBNP levels may have an increased risk of stent-thrombosis and accordingly an increased risk of death:“
See: page 2, introduction, paragraph 3
Results:
“To evaluate the impact of the mode of revascularization on the risk of death in different stages of HF, patients were stratified according to their NT-proBNP median (3042pg/ml).”
See: page 7, results, paragraph 2
Discussion:
“Elevated CRP levels are related to signs of more severe HF such as NYHA classes III or IV or lower LVEF [22] but also to increased risk of death [23].“
See: page 10, discussion, subtitle: prothrombotic state and inflammation, paragraph 3
Conclusion:
“In patients with moderate to severe LVSD and CAD elevated NT-proBNP indicates a prothrombotic state and identifies those patients, in whom the relative risk of PCI is increased compared to CABG.”
See: page 11, conclusion
Statistical analysis.
7. The statistical analysis does not assess the hypothesis that there is an interaction in outcome of interest between those with low or elevated levels of NT-proBNP and PCI or CABG.
A contingency table should be made
NTproBNP | ||
low normal | elevated | |
PCI | mortality | mortality |
CABG | mortality | mortality |
With that model you calculate RR for each group and the SURGICAL PROCEDURE X NTproBNP interaction. That analysis should be able to tell you if the higher mortality in those with greater values of NT-proBNP is different by surgical procedure.
Analyzing the data this way will give you a larger sample size than breaking the analysis into two smaller populations
According to the reviewers suggestions we changed the section “statistical analysis” as follows:
“Various multivariate Cox regression analyses were performed: (I) one model was used in all patients to evaluate the impact of the mode of revascularization on survival and included patient characteristics given in Table 1; (II) to test the hypothesis that there is an interaction in the risk of death between those with lower or elevated NT-proBNP levels and the mode of revascularization (PCI or CABG), the mode of revascularization*NTproBNP interaction as term was forced into a second model. To further test this hypothesis, the relative risk of PCI compared to CABG as reference method was calculated for patients with low and elevatedl NT-proBNP levels using a contingency table. Additional multivariate Cox regression analyses were performed in patients with (III) lower and (IV) elevated NT-proBNP levels as markers of two different stages of disease. To confirm these primary analyses in a more homogenous patient population, the same analyses were repeated using a second data set which excluded patients with single-vessel disease who were treated exclusively with PCI.”
See: page 3, subtitle: statistical analysis,
In the results section we added the following information:
“When the mode of revascularization*NTproBNP interaction as term was forced into a second model, this term was an independent predictor of death (Figure 2B).”
See: page 7, results, paragraph 1
“Relative risk of death of PCI compared to CABG in patients with lower and elevated NT-proBNP
The relative risk of PCI compared to CABG as reference method was similar in patients with lower NT-proBNP - 1.01 (95% confidence interval [CI], 0.45–2.24), but was significantly increased in patients with elevated NT-proBNP – 1.58 (95% CI, 1.07–2.33).”
See: page 7, results, paragraph 5
“Again the mode of revascularization*NTproBNP interaction was an independent predictor of death when forced into a second model (Figure 3B).”
See: page 8, results, paragraph 1
“Relative risk of death of PCI compared to CABG in patients with lower and elevated NT-proBNP
The relative risk of PCI compared to CABG as reference method was similar in patients with lower NT-proBNP - 1.01 (95% confidence interval [CI], 0.45–2.24), but was significantly increased in patients with elevated NT-proBNP – 1.53 (95% CI, 1.05–2.22).”
See: page 8, results, paragraph 4
“Multivariate Cox regression analyses were performed to test the hypothesis that there is an interaction in the risk of death between those with lower or elevated NT-proBNP levels and the mode of revascularization (PCI or CABG). The relative risk of PCI compared to CABG as reference method was calculated for patients with low and elevated NT-proBNP levels.”
See: page 1, abstract, results
8. Are D-dimer values a measure of thrombosis, do you have any of the coagulation parameters for example: INR?
A model with and without D-dimer or any of the coagulation parameters should be performed to assess if thrombosis is a mediator in the association between outcome and levels of NT-proBNP and also perform an interaction model using D-dimer with NT-proBNP and also by surgical procedure.
According to the reviewers suggestions we changed the section “statistical analysis” as follows:
“Bivariate models included NT-proBNP, one of the available parameters of thrombosis (D-Dimer) or coagulation (thromboplastin time, partial thromboplastin time, International Normalized Ratio) and an according interaction term (NT-proBNP*parameter of thrombosis or coagulation) and assessed if thrombosis is a mediator in the association between outcome and levels of NT-proBNP.”
See: page 3, subtitle: statistical analysis,
We added the following parameters to Table 1: thromboplastin time, partial thromboplastin time and International Normalized Ratio.
See: table 1
In the results section we added the following information:
“Using bivariate Cox regression models we found no interactions between NT-proBNP and one of the available parameters for thrombosis (D-Dimer) or coagulation.”
See: page 7, results, paragraph 2
We added the following section to the discussion:
“Etiologies of arterial and venous thrombosis differ. Arterial clots are platelet-rich and formed under high shear stress, whereas venous clots are fibrin-rich and formed during lower shear stress environments. Accordingly, platelets activation is the keystone of athero- and stent-thrombosis. The major causes responsible for platelet activation on long-term after stent implantation include malapposition, neoatherosclerosis, uncovered struts, and stent underexpansion [27]. Platelet activation due to heart failure may potentiate the risk of stent thrombosis. As platelet function testing is not routinely performed, measures for platelet activation are not available in our patients. Available parameters of thrombosis (D-Dimer) or coagulation (thromboplastin time, partial thromboplastin time, International Normalized Ratio) did not demonstrate interactions with NT-proBNP. This may be due to the fact that changes in plasma coagulation are not mainly involved in the development of atherothrombosis. In addition, the number of patients with available parameters of thrombosis and coagulation is limited.”
The new reference (27) were added to the references of the manuscript.
See: page 11, discussion, paragraph 1
Cox regression models including the mode of revascularization and one of the available parameters for thrombosis (D-Dimer) or coagulation also showed no interaction. Due to the small numbers of patients (e.g. 110 patients with D-Dimer) and the small proportion of these patients with CABG (15 patients) we would prefer, not to present these data in the manuscript.
9. Why was a value of 199 NT-proBNP chosen? I assume that the value is ≥ and not just >?
As we took the median of all 398 NT-proBNP values as an cut off to build the two groups, each group consists of 199 patients. Thank you for the hint relating the signs. We changed the description from “>” to “≥” respectively from “above” to “above in all tables, figures and in the manuscript.
10. Line 117 - in this sentence “Those variables, which showed statistically significant influence” substitute “influence” with “association”.
We changed the wording as proposed:
“Those variables, which showed statistically significant association (p-values <0.05) in the univariate model, were further analyzed in a multivariate Cox regression analysis.”
See: page 3, subtitle: statistical analysis,
Results
11. Figures 2 and 3 should be replaced in a different format. The current format makes the forest plots very small and difficult to view. In addition, the authors are repeating information from figure 1. It would be easier to read if figures 2A, 2B and 2C (same for 3A, 3B and 3C) would be shown separately with appropriate titles and figure legends of their own.
We changed Figure 2 and 3 in the way you suggested. Additionally we created another Forest plot including the interaction NT-pBNP*mode of revascularization for all patients (Figure 2B) and for patients with multi-vessel disease (Figure 3B). We also revised Figure 1 to provide more information for the reader.
See: Figure 1, Figure 2 and Figure 3.
12. Which of the surgical procedures is the reference, PCI or CABG?
CABG is the reference procedure. We added this information to the section “statistical analysis” as follows:
“To further test this hypothesis, the relative risk of PCI compared to CABG as reference method was calculated for patients with low and elevated NT-proBNP levels using a contingency table.”
See: page 3, subtitle: statistical analysis,
Moreover, we included this information to the “results” section:
“The relative risk of PCI compared to CABG as reference method was similar in patients with lower NT-proBNP - 1.01 (95% confidence interval [CI], 0.45–2.24), but was significantly increased in patients with elevated NT-proBNP – 1.53 (95% CI, 1.05–2.22).”
See: page 8, results, paragraph 4
Discussion.
13. Line 173: this cannot be stated accurately “Patients with moderate to severe LVSD and CAD who have a lower NT-proBNP level have a long-term survival comparable to CABG after PCI” The appropriate statistical procedures was not done to demonstrate differences in mortality by surgical procedure.
We exchanged this sentence:
“The relative risk of PCI compared to CABG as reference method was similar in patients with lower NT-proBNP, but was significantly increased in patients with elevated NT-proBNP.”
See: page 9, discussion, paragraph 1
Typos or grammar issues
14. Older studies is not very descriptive, please indicate a time range or related to pre or after a certain study was published or procedure was accepted
We changed the manuscript accordingly (introduction):
“Already in the eighties of the last century …”
See: page 2, introduction, paragraph 1
15. Line 51 Insert comma after “recently.”
“Recently, elevated N-terminal proB-51 type natriuretic peptide (NT-proBNP) levels…”
We changed the manuscript accordingly.
See: page 2, introduction, paragraph 2

Reviewer 2 Report
Dear Editor,
It is a privilege for me to serve as review for you estimate JCM. Thank you!
And, I read with great interest the article by Christian Roth et al. with the title: High N-terminal proB-type Natriuretic Peptide Indicates Adverse Outcome After Percutaneous Coronary Intervention Compared to Coronary-Artery Bypass Surgery in Patients With Left Ventricular Dysfunction.
Overall it is a well written manuscript.
MY-MAJOR COMMENT
1) Introduction: need to be more focus on the topic like a short review of the literature about “Natriuretic Peptide” in the study contest.
2) Methods: the nature of the study need to be acknowledged: was your study an observational retrospective study?
3) Statistical methods: Have you calculated from the NT-BNP the sensitivity and specificity… the negative predictive value and the positive predictive value? or its likelihood ratio? ROC and AUC for NT-BNP?
4) Discussion: to my point of view you need to stay focus on the linked literature (NT-BNP and BNP) and explain what is similar and what is different with your results and why. The first part of the discussion is week, but of course it is my opinion.
5) What does this study add to the patient management?
6) What if.... I use and what else?
7) Could your data to be read in the opposite way? A Low NT-BNP is better?
7) Some references:
Maisel AS, Krishnaswamy P, Nowak RM, et al; Breathing Not Properly Multinational Study Investigators. Rapid measurement of B-type natriuretic peptide in the emergency diagnosis of heart failure. N Engl J Med. 2002;347:161-167.
Buse GAL, Koller TA, Burkhart C, Seeberger MD, Filipovic M. The predictive value of preoperative natriuretic peptide concentrations in adult undergoing surgery: a systematic review and meta-analysis. Anesth Analg. 2011;112:1019- 1033.
Vetrugno L. The possible use of preoperative natriuretic peptides for discriminating low versus moderate high surgical risk patoent. DOI: 10.1177/1089253217752061
Bianchi MT, Alexander BM, Cash SS. Incorporating uncertainty into medical decision making: an approach to unexpected test results. Med Decis Making. 2009;29:116-124.
Best Regards,
Author Response
Dear Dr. Zhang,
herewith, we would like to resubmit our manuscript “High N-terminal proB-type Natriuretic Peptide Indicates Elevated Risk of Death After Percutaneous Coronary Intervention Compared to Coronary Artery Bypass Surgery in Patients With Left Ventricular Dysfunction” (Manuscript ID: jcm-491605) following its revision according to the issues raised by the reviewers. In the following, we respond point-by-point to the comments and describe the consequent changes made to the manuscript and the tables. The changes in the manuscript were highlighted in yellow. All authors have read and approved submission of the manuscript. We believe that the adaptations according to the reviewers´ suggestions have significantly improved the manuscript and hope you will concur with this view.
Sincerely yours,
Rudolf Berger
Rudolf Berger, MD, MSc
rudolf.berger@bbeisen.at
Johannes von Gott-Platz 1
7000 Eisenstadt
REVIEWER 2:
MY-MAJOR COMMENT
1) Introduction: need to be more focus on the topic like a short review of the literature about “Natriuretic Peptide” in the study contest.
Our study analyses if NT-proBNP is a mediating factor predicting long-term mortality in patients with LVSD and CAD who have undergone PCI compared to CABG. We clarified this in the “introduction” section as follows:
“Our study analyses if NT-proBNP is a mediating factor predicting mortality in patients with LVSD and CAD who have undergone PCI compared to CABG.”
See: page 2, introduction, paragraph 3
As a prerequisite we shortly described, that natriuretic peptides indicate different stages of disease and predict the risk of death in patients with heart failure. According to the suggestions of the reviewer we inserted the following passage und the recommended reference:
„Natriuretic peptides are well established markers of heart failure [26]. As they vary from normal levels to excessive elevation in patients with moderate to severe LVSD and indicate different stages of disease [11] and predict risk of death [12].“
The new reference (26) were added to the references of the manuscript.
See: page 2, introduction, paragraph 2
After this passage we listed studies showing that elevated natriuretic peptides as markers for a more severe stage of heart failure may also indicate an activated, prothrombotic environment. In order to clarify this message, we inserted the following statement:
“Accordingly, NT-proBNP is positively associated with thrombosis because NT-proBNP is higher in more severe cases of heart failure and, therefore, reflects higher activation of a prothrombotic environment.”
See: page 2, introduction, paragraph 2
2) Methods: the nature of the study needs to be acknowledged: was your study an observational retrospective study?
We added the following information to the abstract and to the “methods” section:
“Methods: This observational retrospective study included …”
See: page 1, abstract, methods
“This single-center observational retrospective cohort study selected …”
See: page 2, methods, paragraph 1
3) Statistical methods: Have you calculated from the NT-BNP the sensitivity and specificity… the negative predictive value and the positive predictive value? or its likelihood ratio? ROC and AUC for NT-BNP?
NT-proBNP is a well established prognostic marker in patients with ischemic cardiomyopathy. The aim of our study was not to test the predictive value of NT-proBNP in this patient population, but to test the hypothesis, that there is an interaction in the risk of death between those patients with low or elevated levels of NT-proBNP and PCI or CABG. We improved the statistical methods to test this hypothesis:
“to test the hypothesis that there is an interaction in the risk of death between those with lower or elevated NT-proBNP levels and the mode of revascularization (PCI or CABG), the mode of revascularization*NT-proBNP interaction as term was forced into a second model. To further test this hypothesis, the relative risk of PCI compared to CABG as reference method was calculated for patients with low and elevated NT-proBNP levels using a contingency table.”
See: page 3, subtitle: statistical analysis,
In the results section we added the following information:
“When the mode of revascularization*NT-proBNP interaction as term was forced into a second model, this term was an independent predictor of death (Figure 2B).”
See: page 7, results, paragraph 1
“Relative risk of death of PCI compared to CABG in patients with lower and elevated NT-proBNP
The relative risk of PCI compared to CABG as reference method was similar in patients with lower NT-proBNP - 1.01 (95% confidence interval [CI], 0.45–2.24), but was significantly increased in patients with elevated NT-proBNP – 1.58 (95% CI, 1.07–2.33).”
See: page 7, results, paragraph 5
“Again the mode of revascularization*NTproBNP interaction was an independent predictor of death when forced into a second model (Figure 3B).”
See: page 8, results, paragraph 1
“Relative risk of death of PCI compared to CABG in patients with lower and elevated NT-proBNP
The relative risk of PCI compared to CABG as reference method was similar in patients with lower NT-proBNP - 1.01 (95% confidence interval [CI], 0.45–2.24), but was significantly increased in patients with elevated NT-proBNP – 1.53 (95% CI, 1.05–2.22).”
See: page 8, results, paragraph 4
4) Discussion: to my point of view you need to stay focus on the linked literature (NT-BNP and BNP) and explain what is similar and what is different with your results and why. The first part of the discussion is week, but of course it is my opinion.
The focus of this work is on the long-term outcome after coronary revascularization and not on the perioperative outcome. During a median observation period of more than four years 44% of these patients died due to their heart failure. We recognize the importance of natriuretic peptides for the assessment of perioperative risk in both cardiac and non-cardiac surgery. In the comparison between interventional and surgical coronary revascularization, however, the periprocedural risk plays a relatively small role in the long-term comparison. Especially with such a long observation time, factors such as a permanently increased platelet activation can lead to different results between these two revascularization methods. Therefore, we believe that the comparison of our data with work dealing with perioperative prognosis makes little sense.
5) What does this study add to the patient management?
We changed the manuscript to clarify possible effects on patient management:
“In patients with moderate to severe LVSD and CAD elevated NT-proBNP indicates a prothrombotic state and identifies those patients, in whom the relative risk of PCI is increased compared to CABG. Previous autopsy-studies suggest that higher rates of stent-thrombosis may account for these results [16,17]. Newer anticoagulants could not improve survival rate in the treatment of patients with LVSD due to CAD: the COMMANDER HF trial failed to reduce mortality when treating patients with a recent exacerbation of HF with reduced systolic function and coexisting CAD with rivaroxaban 2.5 mg bid instead placebo [28]. However, the extension of dual anti-plateled therapy may be a better way to improve survival in these patients [29]. Large outcome studies are needed to test this hypothesis.”
The new references (28 and 29) were added to the references of the manuscript.
See: page 11, conclusions
6) Some references:
Maisel AS, Krishnaswamy P, Nowak RM, et al; Breathing Not Properly Multinational Study Investigators. Rapid measurement of B-type natriuretic peptide in the emergency diagnosis of heart failure. N Engl J Med. 2002;347:161-167.
We added this reference to our manuscript.
See: page 2, introduction, paragraph 2
Buse GAL, Koller TA, Burkhart C, Seeberger MD, Filipovic M. The predictive value of preoperative natriuretic peptide concentrations in adult undergoing surgery: a systematic review and meta-analysis. Anesth Analg. 2011;112:1019- 1033.
Vetrugno L. The possible use of preoperative natriuretic peptides for discriminating low versus moderate high surgical risk patoent. DOI: 10.1177/1089253217752061
Bianchi MT, Alexander BM, Cash SS. Incorporating uncertainty into medical decision making: an approach to unexpected test results. Med Decis Making. 2009;29:116-124.
Please see Point 4 of this revision.

Round 2
Reviewer 1 Report
In the opening line of the introduction the authors say “studies”, but only reference one study. If this was the first study reporting a survival benefit of CABG over medical therapy in patients with poor LVF, then state that and eliminate the sentence “Already in the eighties of the last century”
Proposed sentence “The first study reporting an improved survival benefit of … was published in 1983 by…
Use shorter sentences
Is there evidence that this statement "Accordingly, NT-proBNP is positively associated with thrombosis because NT-proBNP is higher in more severe cases of heart failure and, therefore, reflects higher activation of a prothrombotic environment” is not just an association, but that there is a cause and effect? If so, what is the pathophysiological mechanism by which NT-proBNP stimulates thrombogenesis and include references. My take is that these are two parallel process and that NT-proBNP does not interfere directly with clotting.
This sentence “Acute coronary findings are frequent and usually not clinically diagnosed in HF-patients with coronary artery disease (CAD), in those dying suddenly as well as in those dying from pump failure“ needs reference.
This sentence “These coronary events are thus suggested to be main….” Change to “Coronary events are thus suggested to be main…”. “These” indicate that there are several types of coronary events, but you don’t mention which ones, it is better to leave unspecified.
The hypothesis of this paper is still confusing and requires more step by step organization. It seems that the main gap in knowledge is whether patients with LVSD who require coronary revascularization will have a better survival benefit with CABG or PCI. That seems to be the main point. The second point is if the survival benefit is mediated by NT-proBNP and if NT-proBNP is associated with thrombosis. The authors should write the hypothesis following a sequential order of concatenated ideas
Methods
PCI compared to CABG as reference method, delete the word method. It is the reference outcome.
Eliminate the word “according” from this sentence “an according interaction term (NT-proBNP*parameter of thrombosis or coagulation…”
The cutoff point to differentiate low and high values of NT-proBNP should be stated in the methods section and how was it determined.
NT-proBNP usually has a non-normal distribution and log transformed values are required to use parametric statistics. Report on the distribution of NT-proBNP and proceed accordingly.
What were the criteria why some patients had CABG and others PCI with DES?
What was the median followup time for all subjects? Was the time between CABG or PCI and death ascertained? Are the authors recording death as an acute event after the procedure or anytime between 2004 and 2012?
Results.
Figure 1 contains characters that do not make sense. It is probably a problem caused when uploading the document.
Table 1. What is the median value for NT-proBNP? Add ranges or interquartile range
In table 1, it is not clear which four groups compose table 1. That should be clear from the title in the table and also from the top heading. I assume that it is single vessel + multivessel disease and the other is only those with multivessel disease. That information is not clear from the title.
Table 1 should be divided into at least two other tables. One table could contain information regarding comorbidities, another demographics, and another table with laboratory values.
It is not clear which confounders were included in the model? Are all the variables in Table 1 included as confounders in the model? If this is true, it could lead to over adjustment and type I error.
Figure 2. Is mode of revascularization included in the model? Is the mode of revascularization associated with mortality? The graph does not answer that essential question.
Similar in figure 3.
The model should something like this
Death = age (b1) … + revascularization procedure (b2) + NT-proBNP (b3) + revascularization procedure X NT-proBNP.
Is NT-proBNP included in the model as a categorical variable or a continuous variable?
If you used proc genmod you’ll get relative risk and if you used proc phreg you can have hazard ratios, but you need to calculate followup time for that. Do the authors have a statistician in their team? Did the author estimate relative risks or hazard ratios? In some portions of the results it is written relative risks and others HRs
The authors deviate from their original hypothesis and then included differences in death dependent on the number of vessels affected. If that is one of the gaps in knowledge state that in the introduction. The paper would greatly improve if the authors would focus on their initial hypothesis. The issue of single vessel vs. multivessel can be included in the model as an additional confounder and eliminated from the tables, which causes confusion.
I am concern with the statistical analysis. The fact that on one side the authors mention relative risks and then HRs suggests that the authors did not do the right statistical procedures and for this reason the results cannot be trusted and not the conclusions either.
Once the authors confirm with a statistician and confirm or modify their results, we can resume with the review process and read the discussion and conclusions.
Here is an excellent article on the association between mortality and NT-proBNP. Please include this in the references.
1. Serial measurement of N-terminal pro-B-type natriuretic peptide and cardiac troponin T for cardiovascular disease risk assessment in the Multi-Ethnic Study of Atherosclerosis (MESA). Am Heart J. 2015;170:1170-83.
Author Response
Dear Dr. Zhang,
herewith, we would like to resubmit our manuscript “High N-terminal proB-type Natriuretic Peptide Indicates Elevated Risk of Death After Percutaneous Coronary Intervention Compared to Coronary Artery Bypass Surgery in Patients With Left Ventricular Dysfunction” (Manuscript ID: jcm-491605) following its revision according to the issues raised by the reviewers. In the following, we respond point-by-point to the comments and describe the consequent changes made to the manuscript and the tables. The changes in the manuscript were highlighted in yellow. All authors have read and approved submission of the manuscript. We believe that the adaptations according to the reviewers´ suggestions have significantly improved the manuscript and hope you will concur with this view.
Sincerely yours,
Rudolf Berger
Rudolf Berger, MD, MSc
rudolf.berger@bbeisen.at
Johannes von Gott-Platz 1
7000 Eisenstadt
Reviewer 1:
Comments and Suggestions for Authors
Introduction
1. In the opening line of the introduction the authors say “studies”, but only reference one study. If this was the first study reporting a survival benefit of CABG over medical therapy in patients with poor LVF, then state that and eliminate the sentence “Already in the eighties of the last century”. Proposed sentence “The first study reporting an improved survival benefit of … was published in 1983 by… Use shorter sentences.
We adopted the proposal of the reviewer and shortened the first sentence:
“The first study suggesting a survival benefit of coronary artery bypass graft surgery (CABG) over medical therapy in patients with poor left ventricle function (LVF) was published in 1983 by Alderman and collegues [1]. Recently the STICH trial (Surgical Treatment for Ischemic HF trial) confirmed this survival benefit due to CABG in patients with current guideline-based medical therapy for coronary artery disease and heart failure [2].”
See: page 2, introduction, paragraph 1.
2. Is there evidence that this statement "Accordingly, NT-proBNP is positively associated with thrombosis because NT-proBNP is higher in more severe cases of heart failure and, therefore, reflects higher activation of a prothrombotic environment” is not just an association, but that there is a cause and effect? If so, what is the pathophysiological mechanism by which NT-proBNP stimulates thrombogenesis and include references. My take is that these are two parallel process and that NT-proBNP does not interfere directly with clotting.
To our knowledge, NT-proBNP does not directly interfere with clotting. We clarified this as follows:
“Up to now, there is no evidence that natriuretic peptides interfere directly with clotting. Nevertheless, elevated N-terminal pro B-type natriuretic peptide (NT-proBNP) levels have been associated with spontaneous echo contrast as precursor for left atrial thrombosis [13], as well as with transcatheter heart valve thrombosis [14]. In a large cohort of chronic HF-patients NT-proBNP was an independent predictor of stroke [15]. These findings indicate, that elevated NT-proBNP levels are related with thrombosis. However, this association is probably not based on a direct influence of natriuretic peptides on the coagulation system, but reflects a parallel activation of natriuretic peptides and the coagulation system.”
See: page 2, introduction, paragraph 2.
3. This sentence “Acute coronary findings are frequent and usually not clinically diagnosed in HF-patients with coronary artery disease (CAD), in those dying suddenly as well as in those dying from pump failure“ needs reference.
Reference 16 and 17 already refer to this statement. We have inserted these references after this statement.
“Acute coronary findings are frequent and usually not clinically diagnosed in HF-patients with coronary artery disease (CAD), in those dying suddenly as well as in those dying from pump failure [16, 17].”
See: page 2, introduction, paragraph 3.
4. This sentence “These coronary events are thus suggested to be main….” Change to “Coronary events are thus suggested to be main…”. “These” indicate that there are several types of coronary events, but you don’t mention which ones, it is better to leave unspecified.
We adopted the proposal of the reviewer as follows:
„Coronary events are thus suggested to be main triggers for these two modes of death.“
See: page 2, introduction, pargraph 3.
5. The hypothesis of this paper is still confusing and requires more step by step organization. It seems that the main gap in knowledge is whether patients with LVSD who require coronary revascularization will have a better survival benefit with CABG or PCI. That seems to be the main point. The second point is if the survival benefit is mediated by NT-proBNP and if NT-proBNP is associated with thrombosis. The authors should write the hypothesis following a sequential order of concatenated ideas
We revised the last paragraph of the introduction to make the study hypothesis clearer:
“Stent-implantation during PCI generally involves the risk of stent thrombosis and may significantly increase the risk to suffer further coronary events especially in patients with advanced heart failure and, thereby, activated coagulation system: indeed, the Dutch Stent Thrombosis Registry reported that reduced LVEF is a predictor for stent-thrombosis [18]. These findings suggest that in advanced heart failure (characterized by high NT-proBNP), coronary events clinically perceived as sudden death or death from heart failure are more common in PCI patients than in ACBP patients. In mild stages of heart failure (characterized by low NT-proBNP) where there is no prothrombotic milieu, the frequency of coronary and related clinical events may be equally distributed in PCI and ACBP patients. Our study analyses (1) if patients with reduced LVF who require coronary revascularization will have a better survival benefit with CABG or PCI, and (2) if the survival benefit is mediated by NT-proBNP. “
See: page 2, introduction, paragraph 3.
Methods
6. PCI compared to CABG as reference method, delete the word method. It is the reference outcome.
We adopted the proposal of the reviewer as follows:
“To further test this hypothesis, the relative risk of PCI compared to CABG as reference was calculated for patients with low and elevated NT-proBNP levels using a contingency table.”
See: page 4, methods, paragraph 1.
7. Eliminate the word “according” from this sentence “an according interaction term (NT-proBNP*parameter of thrombosis or coagulation…”
We adopted the proposal of the reviewer as follows:
“Bivariate models included NT-proBNP, one of the available parameters of thrombosis (D-Dimer) or coagulation (thromboplastin time, partial thromboplastin time, International Normalized Ratio) and an interaction term (NT-proBNP*parameter of thrombosis or coagulation) and assessed if thrombosis is a mediator in the association between outcome and levels of NT-proBNP.”
See: page 4, methods, paragraph 1.
8. The cutoff point to differentiate low and high values of NT-proBNP should be stated in the methods section and how was it determined.
The NT-proBNP median (3042 pg/ml; interquartile range: 1313 to 8473 pg/ml) was used as cutoff point to differentiate lower and elevated values. As there was an even number of NT-proBNP values in the data set, the median was found by taking the mean (average) of the two middlemost numbers. We added this information to the methods section (statistical analysis):
“The NT-proBNP median (3042 pg/ml; interquartile range: 1313 to 8473 pg/ml) was used as cutoff point to differentiate lower and elevated values. As there was an even number of NT-proBNP values in the data set, the median was found by taking the mean (average) of the two middlemost numbers.”
See: page 4, methods, paragraph 1.
9. NT-proBNP usually has a non-normal distribution and log transformed values are required to use parametric statistics. Report on the distribution of NT-proBNP and proceed accordingly.
As NT-proBNP levels were not normally distributed, log transformed NT-proBNP levels were used for continuous analysis within the Cox proportional hazards regression models. We added this information to the methods section (statistical analysis):
“As NT-proBNP levels were not normally distributed, log transformed NT-proBNP levels were used for continuous analysis within the Cox proportional hazards regression models.”
See: page 3, methods, paragraph 3.
10. What were the criteria why some patients had CABG and others PCI with DES?
We added the following information to the methods section:
“PCI was usually the preferred option for one – or two-vessel disease not involving the proximal LAD. Especially in patients with complex coronary artery disease (CAD), the decision was mostly reached utilizing a Heart Team approach, which included the opinion of an interventional cardiologist and a cardiovascular surgeon. Often both procedures were possible options and the final decision was that of the patient and his/her advocate physician in the context of the overall clinical presentation. Complex coronary artery disease with a SYNTAX-score (an angiographic tool for grading the complexity of coronary artery disease) >22 was preferably treated by CABG.”
See: page 3, methods, paragraph 1.
11. What was the median follow-up time for all subjects? Was the time between CABG or PCI and death ascertained? Are the authors recording death as an acute event after the procedure or anytime between 2004 and 2012?
The PCI was performed almost exclusively within the hospital stay in which the diagnostic coronary angiography took place. CABG was usually performed within three months after the diagnostic coronary angiography (median waiting time 14 days; IQR 6 to 43 days). The patients were followed up starting with the initial coronary angiography up to the due day (31 December 2014). We added this information to the methods section (study population):
“The PCI was performed almost exclusively within the hospital stay in which the diagnostic coronary angiography took place. CABG was usually performed within three months after the diagnostic coronary angiography (median waiting time 14 days; IQR 6 to 43 days). The patients were followed up starting with the initial coronary angiography up to the due day (31 December 2014).”
See: page 3, methods, paragraph 1.
The survival status including date and type of death was recorded for each patient up to the due date (31 December 2014) with a maximum observation period of up to ten years. On the due date, this survival status was retrieved from the Austrian death registry database. We added this information to the methods section (data collection):
“The survival status including date and type of death was recorded for each patient up to the due date (31 December 2014) with a maximum observation period of up to ten years. On the due date, this survival status was retrieved from the Austrian death registry database.”
See: page 3, methods, paragraph 2.
The median follow-up time for all subjects was already stated in the previous versions of the manuscript:
“During a median observation period of 1,495 days (IQR 819/2,458) 173 patients (44%) died.”)
See: page 4, results, paragraph 2.
Results
12. Figure 1 contains characters that do not make sense. It is probably a problem caused when uploading the document.
This is certainly a bug in the upload. We will check the manuscript carefully after the new upload.
See: Figure 1.
13. Table 1. What is the median value for NT-proBNP? Add ranges or interquartile range
We added the NT-proBNP median and the interquartile range to the title of Table 1. In addition, we added the NT-proBNP median to the column headings.
See: Table 1 and 2.
14. In table 1, it is not clear which four groups compose table 1. That should be clear from the title in the table and also from the top heading. I assume that it is single vessel + multivessel disease and the other is only those with multivessel disease. That information is not clear from the title.
As proposed by the reviewer in point 19, we have excluded the analysis concerning patients with multi-vessel disease. For this reason, the tables is now more clearly arranged.
See: Table 1 and 2.
15. Table 1 should be divided into at least two other tables. One table could contain information regarding comorbidities, another demographics, and another table with laboratory values.
As proposed by the reviewer we divided Table 1 in two other tables - Table 1 contains information regarding demographics, comorbidities and clinical presentation; Table 2 contains baseline therapy and electrocardiographic, echocardiographic and coronary angiographic results. We adapted the titles accordingly.
See: Table 1 and 2.
16. It is not clear which confounders were included in the model? Are all the variables in Table 1 included as confounders in the model? If this is true, it could lead to over adjustment and type I error.
The univariate predictors of death were highlighted in the former Table 1 (now Tables 1 and 2), as indicated in the Methods section (statistical analysis):
„Various Cox proportional hazards regression analyses were calculated: (I) one model included all patients and used univariate significant predictors (patient characteristics marked by *, †, or ‡ in Table 1) to evaluate the impact of the mode of revascularization on survival;“
See: page 3, methods, paragraph 3.
The number of events (deaths) allow to include all univariate predictors:
No. of variables | No. of events (deaths) | |
All | 14 | 156 |
All, including interaction NT-pBNP*mode of revacularization | 15 | 156 |
Above median NT-proBNP | 8 | 119 |
Below median NT-proBNP | 7 | 50 |
We stated this information in the figure legend of Figure 2.
See: Figure 2.
In patients with below median NT-proBNP values, the rule of thumb that Cox models should be used with a minimum of ten outcome events per predictor variable is not exactly fulfilled (50 deaths; six variables). However, this rule has been discussed to be too conservative (Vittinghoff E and McCulloch CE. Relaxing the rule of ten events per variable in logistic and Cox regression. Am J Epidemiol. 2007;165:710-8.). Nevertheless, when excluding the weakest univariate predictor (number of coronary vessels diseased) from the model, the independent predictors stayed the same.
We added this information to the results section (Predictors of death ..):
“The rule of thumb that Cox models should be used with a minimum of ten outcome events per predictor variable is not exactly fulfilled (50 deaths; six variables). However, this rule has been discussed to be too conservative [31]. Nevertheless, when excluding the weakest univariate predictor (number of coronary vessels diseased) from the model, the independent predictors stayed the same.”
See: page 8, results, paragraph 4.
17. Figure 2. Is mode of revascularization included in the model? Is the mode of revascularization associated with mortality? The graph does not answer that essential question. The model should something like this:
Death = age (b1) … + revascularization procedure (b2) + NT-proBNP (b3) + revascularization procedure X NT-proBNP.
As proposed by the reviewer we adapted the figure legend of Figure 2 as follows:
“ Figure 2. Cox proportional hazards regression analyses for all-cause mortality in (A) all patients (156 deaths; Death = age (b1) + gender (b2) + cerebrovascular disease (b3) + peripheral vascular disease (b4) + prior myocardial infarction (b5) + diabetes mellitus (b6) + creatine (b7) + rhythm (b8) + NYHA (b9) + CCS (b10) + systolic left ventricular function (b11) + total cholesterol (b12) + number of coronary vessels diseased (b13) + NT-proBNP (b14)), (B) all patients including interaction NT-pBNP*mode of revacularization (b15), (C) patients with above median NT-proBNP (119 deaths; Death = age (b1) + gender (b2) + peripheral vascular disease (b3) + prior myocardial infarction (b4) + prior CABG (b5) + creatine (b6) + mode of revascularization (b7) + NT-proBNP (b8)), (D) patients with below median NT-proBNP (50 deaths; Death = age (b1) + gender (b2) + cerebrovascular disease (b3) + diabetes mellitus (b4) + creatine (b5) + number of coronary vessels diseased (b6) + NT-proBNP (b7)). “
See: Figure 2.
18. Is NT-proBNP included in the model as a categorical variable or a continuous variable?
Please see point 9.
19. The authors deviate from their original hypothesis and then included differences in death dependent on the number of vessels affected. If that is one of the gaps in knowledge state that in the introduction. The paper would greatly improve if the authors would focus on their initial hypothesis. The issue of single vessel vs. multivessel can be included in the model as an additional confounder and eliminated from the tables, which causes confusion.
As proposed by the reviewer, we have excluded the analysis concerning patients with multi-vessel disease. In the Cox model the issue of single vessel versus multivessel coronary artery disease had no impact on survival. We added this information to the discussion (limitations):
„However, in the Cox model the issue of single vessel versus multivessel coronary artery disease had no impact on survival”
See: page 11, discussion, paragraph 4.
20. If you used proc genmod you’ll get relative risk and if you used proc phreg you can have hazard ratios, but you need to calculate followup time for that. Do the authors have a statistician in their team? Did the author estimate relative risks or hazard ratios? In some portions of the results it is written relative risks and others HRs. I am concern with the statistical analysis. The fact that on one side the authors mention relative risks and then HRs suggests that the authors did not do the right statistical procedures and for this reason the results cannot be trusted and not the conclusions either. Once the authors confirm with a statistician and confirm or modify their results, we can resume with the review process and read the discussion and conclusions.
At the time of first submission our manuscript only included Cox proportional hazards regression models. We included this detailed information in the statistical section (“Cox proportional hazards regression analyses were performed to evaluate predictors of death.”). Of course, the calculated hazard ratios take account not only of the total number of events, but also of the timing of each event. Due to the suggestions of the first reviewer
(“A contingency table should be made
NTproBNP | ||
low normal | elevated | |
PCI | mortality | mortality |
CABG | mortality | mortality |
With that model you calculate RR for each group …“)
we further included the relative risk of PCI compared to CABG as reference, which was calculated for patients with low and elevated NT-proBNP levels using a contingency table. Our statistician Michael Weber, PhD confirmed the correctness of these calculations (https://www.researchgate.net/profile/Michael_Weber9; michael.weber@meduniwien.ac.at).
We honored his statistical support in the acknowledgments.
See: page 9, results, paragraph 2 and acknowledgments.
21. Here is an excellent article on the association between mortality and NT-proBNP. Please include this in the references.
Serial measurement of N-terminal pro-B-type natriuretic peptide and cardiac troponin T for cardiovascular disease risk assessment in the Multi-Ethnic Study of Atherosclerosis (MESA). Am Heart J. 2015;170:1170-83.
We added this article to the introduction section (paragraph 2):
“Natriuretic peptides are well established markers of LVSD HF [30] and HF [26].”
See: page 2, introduction, paragraph 2.

Reviewer 2 Report
Thank you for the opportunity to read your revised manuscript.
The work is improved a lot. I m satisfy.
Author Response
No comments.
Round 3
Reviewer 1 Report
This manuscript is progressing nicely, but still requires significant changes. The information reported by the authors is of significant clinical importance.
Define ACBP or delete it and use CABG
IN line 83, change mediated to predicted. “If the survival benefit is predicted”, not mediated, by NT-proBNP. As the hypothesis is more clear now, predicted is a better term.
Table 1 add ≤ or ≥ symbols where appropriate
Table 3, remove unnecessary abbreviations
In figure 2
Sorry I confused the authors with the equation, but there is no need to add B1…Bn in the model, just list the covariates that were used for adjustment. Usually, the covariates are listed and explained in the methods section.
The Cox proportional hazard models need to be better organized to represent the hypothesis its assessing. Some of the covariates used in the model are related to each other and you only need to use one in the model. For example, using NYHA class and systolic left ventricular function, which I assume is ejection fraction? Or is it cardiac index? Do you have eGFR? That would be better than serum creatinine. The variables included in the model should be variables that influences both the dependent variable and independent variable, causing a spurious association, confounders. The confounders should be maintained the same between models, except for what is explained below.
To answer if there is an association between type of revascularization procedure and death:
death adjusted by age, sex… other confounders and revascularization procedure
To answer if death is associated by levels of NT-proBNP than
death adjusted by age, sex… other confounders and NT-proBNP
To answer if there is an interaction between revascularization procedure and NT-proBNP, it is recommended that NT-proBNP is used as a categorical variable (NT-proBNP above or below the median
death adjusted by age, sex… other confounders, NT-proBNP categorial, revascularization procedure, NT-proBNP categorialXrevascularization procedure.
This model should provide a statistical answer to what is described in table 3, that the unadjusted proportion of death is higher in those getting PCI when compared to CABG, but only in those with levels of NT-proBNP above the median.
By doing the interaction term you don’t need to do models C and D, but if you use models C and D, you don’t have to add NT-proBNP in the model, as you are already adjusting for NT-proBNP by excluding patients with NT-proBNP above or below the median.
For clinical purposes it is not only important to say that NT-proBNP is associated with greater mortality following PCI or CABG, but to define a cutoff point, in that sense, instead of using NT-proBNP as a continuous variable, I recommend that NT-proBNP is included as a categorial variable i.e., above and below the median, as described in this paper.
In the results section line 232 where it says “The relative risk of PCI compared to CABG as reference method was similar in patients with lower” It should say “The relative risk of death in patients having a PCI compared to CABG was similar in patients with lower…”
A table is needed to show if the patients who were chosen to have PCI have more or less risks factors for mortality from CVD than those chosen to have CABG.
The statement regarding the rule of thumb for Cox proportional hazard model should be in the statistical section.
Discussion.
I would say that death is associated to the mode of revascularization, but only in those with levels of NT-proBNP above the median.
This statement “In contrast, this registry found no difference in long-term survival between PCI compared to CABG [6].” Underestimates the main finding of this analysis, which is that there is a difference in HR for death by mode of revascularization, but only at elevated levels of NT-proBNP.
The discussion regarding a prothrombotic state is overstated as this analysis does not focused on that area and the only mention of coagulation in the results section is in table 2, but there are no further statistical analysis measuring an association between thrombotic state and death. The authors should perform more calculation to determine the influence of any of the measures of coagulation with mortality and the association with revascularization procedure to be able to conclude “In patients with moderate to severe LVSD and CAD elevated NT-proBNP indicates a prothrombotic state and an increased risk of death after PCI compared to CABG”. If the authors eliminate the addressing “prothrombotic state” and focus on the findings, then there is no need for further statistical analysis.
The discussion on inflammation and prothrombotic state is too speculative and I recommend that it is shorten and more directly discusses and explains the associations found in this study.
Round 4
Reviewer 1 Report
This manuscript has greatly improved and it is almost ready.
Statistical section:
It is standard to classify statistical models by numbers and not letters. If the list of confounders is similar between models, one can say: model 1 + additional confounders”. No need to repeat the same confounders in different models. For example if model 1 = age, sex and race and model 2 adds NT-proBNP, than you can write model 2 = model 1 + NT-proBNP.
Results
Line 171 where it says “(IQR 819/2,458)” eliminate the / and add – to separate the two numbers.
Line 245 and 246, the authors are calculating HR, not relative risks, change to HRs.
In line 247 and 248, it appears that there is a negative sign before the HR, delete the line.
Discussion
In line 251 and 252, that authors state the death is “… only in those patients with reduced LVF and elevated NT-proBNP…”, eliminate LVF, as I understand they all have low LVF, NT-proBNP was the discriminating factor.
In this sentence “NT-proBNP is a predicting factor for the revascularization procedure..:” NTproBNP is a predicting factor for what? Include “death” in the sentence.
In this sentence “There is a lack of randomized controlled trials testing PCI versus medical therapy in patients with LVSD.” Please add testing for what? Please be specific what would be the primary outcome if you were to design a clinical trial?
Line 259, add a coma after recently “Recently, a large meta-analysis…”
In the paragraph regarding mode of death it is important to mention the median time between procedure and death. Did most death occurred during the procedure of sometime after? This paragraph is too long and it should focus more on the results from the study. The timing of death is an important issue to discuss further because if death occurred during the procedure than it could be associated with the procedure than when it occurs later after the procedure. The issue of underreporting or misreporting is an important one, but it should not take more than a couple of sentences to highlight the issue.
Explain with further detail, why this statement “severe HF patients treated with PCI had a worse prognosis than those treated with CABG”, indicates that death by CAD is underreported?
Change the word “depends” in this sentence “…activation in HF depends on the severity of the disease as…” to “is associated with” depends implies cause and effect, and this manuscript does not show that.
Although platelet activation is not measure, is the use of aspiring and its platelet clotting abilities a way to assess the effect of platelet activation in these cases? The author should elucubrate on that, briefly, if there is any available evidence.
The authors should stick to the available evidence found in this study. Delete this sentence from the conclusion as this information is not available from the results of this study “Newer anticoagulants could not improve survival rate in the treatment of patients with LVSD due to CAD [24].” It is appropriate to postulate future studies based on gaps in knowledge that are left after this study.